# Effects of Vitamin D in Post-Exercise Muscle Recovery. A Systematic Review and Meta-Analysis

**DOI:** 10.3390/nu13114013

**Published:** 2021-11-10

**Authors:** Hugo J. Bello, Alberto Caballero-García, Daniel Pérez-Valdecantos, Enrique Roche, David C. Noriega, Alfredo Córdova-Martínez

**Affiliations:** 1Department of Mathematics, School of Forestry Industry and Agronomic Engineering and Bioenergy, GIR: “Physical Exercise and Aging”, University of Valladolid, Campus Universitario “Los Pajaritos”, 42004 Soria, Spain; hugojose.bello@uva.es; 2Department of Anatomy and Radiology, Health Sciences Faculty, GIR: “Physical Exercise and Aging”, University of Valladolid, Campus Universitario “Los Pajaritos”, 42004 Soria, Spain; albeto.caballero@uva.es; 3Department of Biochemistry, Molecular Biology and Physiology, Health Sciences Faculty, GIR: “Physical Exercise and Aging”, University of Valladolid, Campus Universitario “Los Pajaritos”, 42004 Soria, Spain; danielperezvaldecantos@gmail.com; 4Department of Applied Biology-Nutrition, Institute of Bioengineering, University Miguel Hernández, 03202 Elche, Spain; eroche@umh.es; 5CIBER Fisiopatología de la Obesidad y Nutrición (CIBEROBN), Instituto de Salud Carlos III (ISCIII), 28029 Madrid, Spain; 6Alicante Institute for Health and Biomedical Research (ISABIAL), 03010 Alicante, Spain; 7Department of Surgery, Ophthalmology, Otorhinolaryngology and Physiotherapy, Faculty of Medicine, Hospital Clínico Universitario de Valladolid, 47005 Valladolid, Spain; dcnoriega1970@gmail.com

**Keywords:** creatine kinase, exercise, muscle damage, myoglobin, recovery, vitamin D

## Abstract

Vitamin D is a key micronutrient modulating function and health in skeletal muscle. Therefore, we sought to systematically review the role of vitamin D in muscle recovery. A search in different databases (PubMed/MEDLINE, WOS, Google Scholar, and Scopus) was carried out following PRISMA^®^ and PICOS. The search period was from inception to April 2020. Changes in post-exercise muscle damage were quantified comparing experimental group vs. placebo in each study by using number of participants, standardized mean difference (SMD), and standard error of the SMD. Hedges’s g was used to calculate the SMDs for each study group and biased by the inverse of variance that allows calculating an overall effect and the 95% confidence interval (CI). The net vitamin D supplementation effect was calculated by subtracting the placebo SMD from SMD of the experimental group. The DerSimonian and Laird method was used as a random effect model, taking into account that the effect of vitamin D on muscular damage may vary according to the dose administered and additional moderators. Six studies were selected. In conclusion, regarding circulating levels of muscle biomarkers and additional limitations of the studies, it cannot be concluded that vitamin D supplementation exerts an effect in post-exercise muscle recovery. Likely, the anti-inflammatory action of vitamin D is quicker than the recovery of tissue structure and function. This aspect is pending verification in future research.

## 1. Introduction

Vitamin D is a micronutrient that modulates different pathways related to immune function, including mitochondrial dysfunction (due to oxidative stress and reactive oxygen species production), autophagy, and inflammation [1,2]. In this context, vitamin D is involved in the differentiation of monocytes to macrophages, increasing chemotaxis and phagocytosis, among others. This results in the prevention of immunopathological processes [3]. In addition, vitamin D is involved in the modulation of inflammatory processes which affect muscle function, by stimulating protein synthesis and cellular growth [4,5,6,7].

In this context, an association between myopathy and deficient levels of vitamin D has been documented. Muscle pathology presents deterioration of muscle fibers that results in muscle atrophy. Affected people undergo weakness, endurance decline, chronic inflammation, and penetration of inflammatory cells in muscular tissue, together with low circulating levels of 25-hydroxy-vitamin D (25(OH)D) [7,8]. Therefore, the administration of 25(OH)D to patients with certain muscular pathologies can improve strength and skeletal muscle function [9]. In addition, Barker et al. revealed that adequate circulating levels of 25(OH)D could reduce weakness after muscle damage due to intense exercise in non-supplemented subjects [10]. These results could be interpreted as a change in the ratio of proliferation and apoptosis in favor of the first one, rather to variations in the amount of infiltrating leukocytes or satellite cells [11,12].

Exercise-induced muscle damage (EIMD) results from the disruption of skeletal myocytes and the subsequent release of intracellular muscle proteins in blood [13,14,15]. Clinical consequences include strength loss, inflammation, impaired motion range, and delayed onset of muscle soreness [13]. Creatine kinase (CK), myoglobin (Mb), and lactate dehydrogenase (LDH) are the most representative biomarkers of muscle damage [14,15,16,17,18]. From all these biomarkers, CK is the most specific and commonly used to assess the harshness of muscle damage [14,15,16,17]. CK reaches the blood via increased membrane permeability that occurs after exercise execution. In addition, plasma CK amounts are conditioned by many factors related to exercise, such as intensity, duration, training status of the individual, or exercise experience [14,15]. The amount of muscle mass has to be taken also into account which depends on genetics and gender [14,15,16,17,19]. Nonetheless, muscle damage displays a huge inter-individual variability depending on additional factors that could exert an additional influence [13,20,21].

Paulsen et al. proposed different levels (mild, moderate, and severe) for muscle damage regarding the reduction in force-generating capacity (FGC) and CK activity in serum [12]. High serum levels of muscle damage biomarkers (CK, LDH, and Mb) occur after the execution of sustained and intense exercises. These increases go in parallel with increases in catabolic hormones, such as cortisol, favoring protein degradation processes in muscle [4,5,6,7,12,22]. In addition, increased muscle damage results in the accumulation of oxidized inactive end-products (i.e., protein carbonyls, malondialhehyde, and oxoguanine, among others), affecting performance in a negative way and delaying the required time for optimal recovery [14,16,22,23,24,25].

In this line, Beaudart et al. have documented that low serum levels of vitamin D occur with altered muscle metabolic response and decreased strength [26]. Furthermore, the accumulated evidence indicates that low levels of vitamin D are related to muscle weakness, pain, impaired balance, and increased risk of lesions in elderly [6,7]. Altogether, scientific findings support a significant effect of optimal levels of vitamin D on performance and damage prevention. In this context, clinical studies have documented muscle strength improvements with vitamin D supplementation [27]. In rats, vitamin D reduced the release in plasma of CK after a demanding exercise test [28]. Inhibition of apoptosis and an increased synthesis of extracellular matrix proteins are proposed mechanisms to explain an enhanced recovery in maximal isometric strength after vitamin D supplementation [11].

In this context, a meta-analysis by Tomlinson et al. explored the result of vitamin D supplementation on muscle strength [29]. The authors found that vitamin D supplementation resulted in a positive outcome on strength indices on upper and lower limbs. From this meta-analysis, it can be inferred that administration of vitamin D3 every day would be more effective than administration once a week or monthly in order to maintain circulating vitamin D levels [29].

Despite the well-documented benefit of vitamin D supplementation in reducing muscular exercise-associated inflammation, only few experimental studies demonstrated an improvement in muscle recovery. Therefore, the main aim of the present systematic review and meta-analysis is to analyze if the accumulated scientific evidence supports an instrumental role for vitamin D in muscle recovery from data presented by selected articles published.

## 2. Materials and Methods

### 2.1. Searching Strategies

The present report is a systematic review with a meta-analysis aiming on the effect of vitamin D on recovery from post-exercise muscular damage. Recovery does not include a reduction in bone fractures which has been a very controversial field of research (https://www.saragironicarnevale.com/science-magazine-tide-of-lies, accessed on 1 August 2021). A review process was carried out according to the guidelines of Preferred Reporting Items for Systematic review and Meta-Analyses (PRISMA), assisting to increase into the integrity of this report [30]. 

To determine the inclusion criteria, we used the PICOS model [31]. P (Population): “athletes”; I (Intervention): “Vitamin D supplementation”; C (Comparators): “same conditions in placebo”; O (Outcome): “CK/LDH/Mb levels after exercise”; and S (Study design): “clinical trial”. A well-designed search was focused in the following databases: PubMed/MEDLINE, web of science (WOS), Google Scholar, and Scopus, including results until 1 April 2020. 

A combination of medical subject headings (MeSH) and free-text terms for key concepts in relation to vitamin D supplementation and muscular damage were included. The following search equation was used: (“Vitamin D” [All Fields] OR “calciferol” [All Fields]) AND (“Muscle Damage” [All Fields] OR “Creatine kinase” [All Fields] OR “CK” [All Fields] OR “Lactate” [All Fields] OR “LDH” [All Fields]“ OR “Mb” [All Fields]“). The snowball strategy resulted in relevant articles in the field. Cross-references from titles and abstracts were used to identify duplicate and missing studies. We screened titles and abstracts for a later full-text revision. 

### 2.2. Inclusion and Exclusion Criteria

We want to point out that no filters were applied to the performance level of athletes, gender, race, or age, in order to increase the analysis power. The following inclusion criteria were applied as filters to the articles obtained from the search: (1) the presence of an experimental group supplemented with vitamin D before and/or during exercise performance compared to an identical experimental placebo group; (2) the effects of Vitamin D on exercise tests and/or in real/simulated competitions; (3) randomized design; (4) information regarding vitamin D administration (dose and time); and (5) published in languages different from English. The following exclusion criteria were applied: (1) studies not performed in athletes; (2) studies performed into a therapy protocol; and (3) no placebo/control group. We consider the term athletes as all sportsmen and sportswomen that perform a planned physical activity regarding a certain intensity, volume, and frequency. 

### 2.3. Data Extraction

We applied the inclusion and exclusion criteria on each study and we extracted the following data using a spreadsheet: study source (authors and year of publication), study design, vitamin D supplementation (dose and timing), sample size, participant characteristics (level and gender), and final intervention outcomes. Experiments were clustered by the type of marker used to evaluate muscular damage (CK, LDH or Mb). 

Data regarding mean, standard deviation (SD), and sample size were obtained from the tables and figures of the selected publications. When necessary, we communicated with the authors of publications to obtain additional data. When it was not possible, we extrapolated the data from the figures.

### 2.4. Potential Bias Risk

Table 1 summarizes the bias risk in the experimental design of the 6 selected publications [32,33,34,35,36,37]. All the studies, except for one, are randomized and blinded. We believe that the experiments have an acceptable quality and that the meta-analysis is not affected by any bias. In addition, we believe that placebo and experimental groups were well matched in the majority of the studies. The only concerns in this regard could be Żebrowska et al. [37], since their study was not randomized, and Parsaie et al., since they reported differences in baseline body mass index (BMI) between the groups [34].

### 2.5. Statistical Analysis

Data from participants were stated as mean ± SD. Descriptive analyses and bias risk were achieved using a spreadsheet (Microsoft Excel-2016©). The R programming language was used for meta-analytic statistics. The number of participants, standardized mean difference (SMD), and standard error of the SMD were used to quantify changes in variables related to muscle damage when comparing experimental vs. placebo groups. SMDs for each study group were calculated using Hedges’s g and weighted by the inverse of variance to calculate an overall effect and the 95% confidence interval (CI) [38]. The effect of vitamin D supplementation was obtained by subtracting the control/placebo SMD from the SMD of the experimental group. Taking into account that the effect of vitamin D on recovery from muscular damage may differ depending on dose and other moderators related to participants, we decided to use a random effects model with the DerSimonian and Laird method (DerSimonian and Laird, n.d.) [39]. 

The *I^2^* statistics were calculated to assess heterogeneity (systematic differences) across studies. This parameter indicates the total variation percentage observed across studies, due to real heterogeneity rather than casual [40]. *I^2^* interpretation is intuitive and scores between 0–100%. *I^2^* values between 25–50% represent a small amount of inconsistency. *I^2^* values between 50–75% indicate a medium amount of heterogeneity. Finally, an *I^2^* value > 75% signifies a large amount of heterogeneity [41]. A restrictive categorization for *I^2^* would not be appropriate for all circumstances, although it would accept terms, such as low, moderate, and high for *I^2^* values of 25%, 50%, and 75%, respectively.

## 3. Results

### 3.1. Article Search

Searches in the selected databases identified initially 782 publications related descriptors. However, only six articles matched with all the inclusion criteria (Figure 1). From the 782 initial articles, 32 were duplicates and, thereby, eliminated. From the remaining 750, 682 were excluded because they were not clinical trials. From the 65 remaining articles, 59 were removed because they were not made in humans, they were not placebo-controlled, or they did not measure muscular damage.

### 3.2. Vitamin D Administration Protocols

The main characteristics of the interventions are included in Table 2. The amount and time of vitamin D supplementation are presented in Table 3. The study of Pilch et al. (2020) [36] was subdivided in four groups. In this study [36], participants were divided at the beginning of the study in two groups: participants with optimal levels (Figure 2, Figure 3 and Figure 4) of circulating vitamin D and participants with suboptimal levels of vitamin D (Figure 2, Figure 3 and Figure 4). Then, these groups were divided into two groups: experimental or supplemented (Figure 2, Figure 3 and Figure 4) and control or non-supplemented (Figure 2, Figure 3 and Figure 4). All groups performed a test at the beginning (t = 0 in Figure 2, Figure 3 and Figure 4) and the same test at the end of intervention (t = 3 months in Figure 2, Figure 3 and Figure 4). The test consisted of a treadmill inclined at 10% and adjusted to 60% of VO_2_peak. To induce muscle damage, participants carried and extra-weight corresponding to 5% of their body weight [36]. Sample size and outcomes of the studies are shown in Table 4. The total sample analyzed from all selected reports was 198 athletes.

Regarding the exercise probes performed in the selected studies, NASCAR pit crew are well-trained people in all physical qualities, requiring strength, endurance, speed, and coordination [32]. In the work of Shanely et al., participants performed a variety of sports, but they executed similar activities during intervention [33]. On the other hand, participants from the work of Parsaie et al. [34] trained soccer. Soccer training is very standardized and the physical tests to control physical skills are also very homogeneous. Therefore, results from the study are optimal to be compared with others. Ashtary-Larky et al. studied 14 women training regularly in an endurance physical activity (3 d/week) at least 2 years before the intervention [35]. Therefore, the physical activity performed is standardized for endurance sports. However, the authors did not indicate the exact type of activity. As described before, Pilch et al. studied 60 physically active volunteers divided into two groups at the beginning of intervention [36]. Although the proposed evaluation tests are adequate, the authors do not indicate the type of physical activity performed during the intervention. They only indicated that activity was performed a low intensity (1385 ± 116 MET-min/week, on average). Furthermore, the activity control was carried out by using the International Physical Activity Questionnaire (IPAQ). This is a very subjective evaluation, since each participant filled his own questionnaire with no supervision. Finally, Żebrowska et al. studied 24 runners, who were all competitors of ultramarathons during the National Running Championships [37]. In this particular study, the physical activity performed by the athletes is regulated and well controlled.

### 3.3. Effect of Vitamin D Supplementation on Circulating Creatine Kinase Levels

Vitamin D produces no significant effect on CK levels after exercise (SMD, −0.10; 95% CI, −0.70 to 0.51; I2, 72%; random effects *p* = 0.7225) (Figure 2). Figure 2 shows the forest plot comparing the effects of vitamin D supplementation on CK. The first column of the plot contains all the studies (Table 4). The study of Pilch et al. was included several times, as indicated before [36].

### 3.4. Effect of Vitamin D Supplementation on Circulating Lactate Dehydrogenase Levels

Vitamin D produces no significant effect on LDH after exercise (SMD, −0.06; 95% CI, −0.69, to 0.58; I2, 71%; random effects *p* = 0.7987) (Figure 3). Figure 3 shows the forest plot comparing the effects of vitamin D supplementation on LDH. The first column of the plot contains all the studies (Table 4). The study by Pilch et al., was included several times, as indicated in Section 3.2 [36].

### 3.5. Effect of Vitamin D Supplementation on Circulating Myoglobin Levels

Vitamin D produces no significant effect on Mb after exercise (SMD, 0.08; 95% CI, −1.08, to 1.24; I2, 71%; random effects *p* = 0.8779) (Figure 4). Figure 4 shows the forest plot comparing the effects of vitamin D supplementation on Mb. The first column of the plot contains all the studies (Table 4). The study Pilch et al. was included several times, as indicated in Section 3.2 [36].

## 4. Discussion

Vitamin D affects muscle function and structure. This implies that optimization of circulating vitamin D levels and supplementation can result in improved physical capacity in healthy individuals. For instance, Close et al., showed an improvement in vertical jump height and sprint time after vitamin D supplementation in athletes with low circulating levels of vitamin D [42]. However, Owens et al. reported no effect of vitamin D supplementation on strength [43]. In a subsequent report, the same research group published that vitamin D supplementation (6 weeks, 4000 IU/day) improved peak torque recovery after eccentric exercise in vitamin D deficient men [44]. These discrepancies may suggest that the basal circulating vitamin D levels of athletes seem to be instrumental in the process of adaptation to intense exercise.

Altogether, the few published results regarding muscle recovery, presented in this meta-analysis seem to indicate that vitamin D supplementation does not exert a significant effect on post-exercise muscle recovery. Nevertheless, other studies indicate that the main part of injuries that undergo swimmers and divers are coincident with low circulating levels of vitamin D [45]. In the same line, Lewis et al., have indicated that low levels of circulating vitamin D are linked to increased muscle weakness and fatigue [46]. In this context, we can hypothesize that fatigue reduction could shorten recovery time. This could be related to a modulation of the inflammatory component based on the expression of specific cytokines. In this regard, Barker et al. reported that vitamin D supplementation diminished the release of inflammatory messengers after high-intensity eccentric and concentric actions [47].

Therefore, the main limitation of the studies presented in this meta-analysis is that the anti-inflammatory action exerted by vitamin D is not clearly monitored by only analysing the circulating levels of specific muscle markers. Otherwise said, the efficiency of vitamin D supplementation decreasing immunological markers of inflammation does not seem to go in parallel with a decrease in circulating biomarkers of muscle damage. To circumvent this problem, cytokine determinations should be included in studies regarding muscle recovery. In this line, parameters determined in regular blood analysis do not seem to be useful to monitor muscle recovery after vitamin D supplementation. In addition, differences in baseline 25-hydroxyvitamin D concentrations before supplementation (likely due to skin biosynthesis), as well as to doses and times of administration, should be taken as well into account [42]. In this context, Ashtary-Larky et al., used a megadose administered once and no changes in circulating muscle markers were reported [35]. This observation may suggest that administration of vitamin D every day would be more effective than administration once a week or monthly [44]. In addition, the most optimal dose and duration of supplementation is a pending question for future research. Altogether, these points could explain certain discrepancies that exit in the published reports regarding the role of vitamin D in post-exercise recovery. Additional research in necessary in this respect.

Few studies obtained positive recovery starting with optimal serum levels of vitamin D, except for Parsaie et al., Ashtary-Larky et al., and a subgroup of Pilch et al. [34,35,36]. Furthermore, Nieman et al. [32] found no changes in muscle biomarkers with an optimal dose of 3800 IU of vitamin D/day for 6 weeks. This particular study was performed with participants working in the pitline of the NASCAR tests. The subjects only performed 90 min of an eccentric exercise not characterized appropriately. The authors observed that the circulating levels of muscle enzyme biomarkers were higher after exercise in the vitamin D-supplemented group. In our opinion, this result could be interpreted taking into account the influence of external factors, such as motivation, lifestyle (uncontrolled), nutritional habits, etc. We believe that the results shown are not relevant for understanding the effect of vitamin D on muscular damage biomarkers. Finally, Żebrowska et al. have communicated that, after 1 h, Mb and CK levels were lower in athletes after vitamin D supplementation without changes in LDH activity [37]. The authors concluded that a vitamin D response displayed a negative correlation to muscle damage biomarkers; this effect was more marked in the subsequent 24 h of the recovery phase. They concluded that 3 weeks of supplementation results in optimal effects for few parameters related to muscular function.

In addition, the analytical methodology to determine circulating levels of vitamin D may explain some contradictory results in the reports of this meta-analysis. In fact, liquid chromatography–mass spectrometry is the gold standard for measuring vitamin D, performed only in [32]. On the other hand, immunoassay seems to have a lower specificity, performed in [33,34,35,36]. Finally, vitamin D3 (cholecalciferol) seems to be converted more specifically in active vitamin D than vitamin D2 (ergocalciferol). This efficacy seems to be related to the higher affinity of cholecalciferol for the hydroxylation enzymes in liver and kidney [48]. Altogether, and according to the I2 values, all studies selected in this meta-analysis present a high degree of heterogeneity. Limitations of the selected studies are indicated in Table 5. This could be due to the different physical probes used in these studies: concentric, eccentric, aerobic, among others. A consensus protocol or exercise routine will offer more comparable results.

## 5. Conclusions

This is the first meta-analysis analyzing published reports studying the effect of vitamin D supplementation on muscle damage biomarkers as indicators of post-exercise recovery. Although vitamin D seems to be effective against the muscular inflammatory process, as we have proposed in a previous report, the role in post-exercise recovery by modulating the release of muscle biomarkers remains to be demonstrated [49]. We could suggest that, to include cytokine determinations, a longer administration time, or higher doses of supplementation would be variables to take into account for future research. In addition, skin biosynthesis after sun exposure could be considered an additional variable to take in account when working with a control non-supplemented group. From a practical point of view, vitamin D supplementation serves to normalize and optimize its own blood levels. However, additional studies with comparable protocols are necessary to reach more solid evidences regarding post-exercise muscle recovery.

## Figures and Tables

**Figure 1 nutrients-13-04013-f001:**
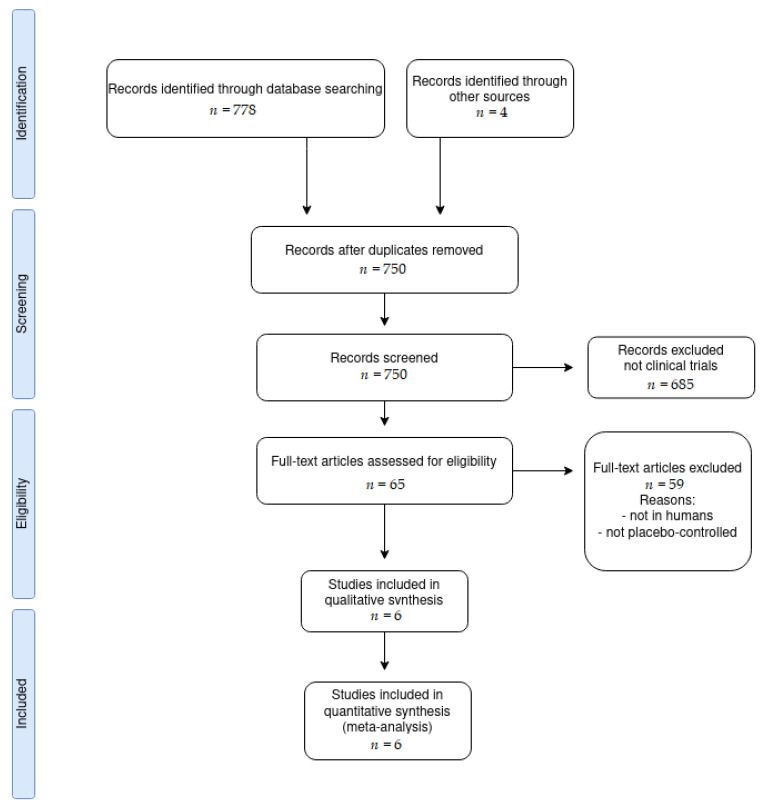
Flow chart of the process of study selection.

**Figure 2 nutrients-13-04013-f002:**
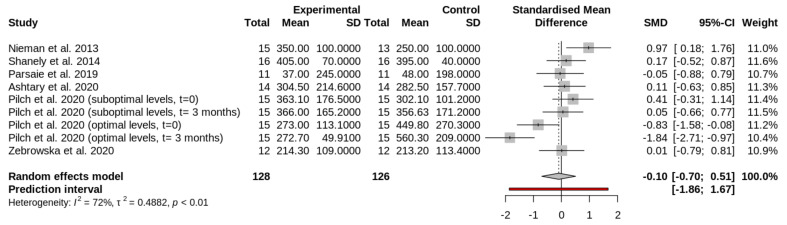
Forest plot comparing the effects of vitamin D supplementation on circulating CK levels after exercise. The squares in the forest plot on the left side of the line represent studies or groups in which the mean of the experimental group (supplemented) was lower than the mean of the control group (placebo). The ones in the right side indicate the opposite.

**Figure 3 nutrients-13-04013-f003:**
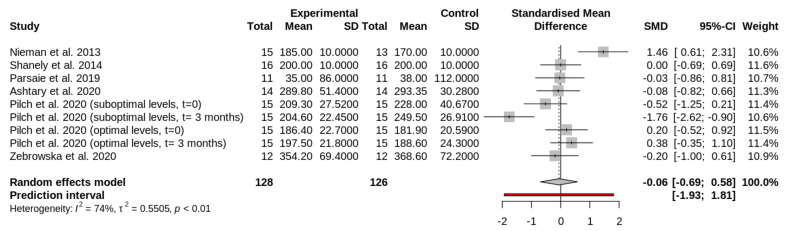
Forest plot comparing the effects of vitamin D supplementation on circulating LDH levels after exercise. The squares in the forest plot on the left side of the line represent studies or groups in which the mean of the experimental group (supplemented) was lower than the mean of the control group (placebo). The ones in the right side indicate the opposite.

**Figure 4 nutrients-13-04013-f004:**
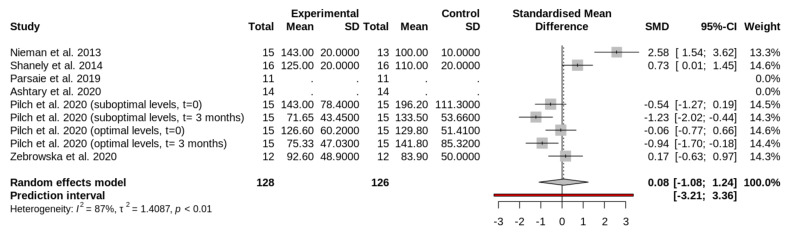
Forest plot comparing the effects of vitamin D supplementation on circulating Mb levels after exercise. The squares in the forest plot on the left side of the line represent studies or groups in which the mean of the experimental group (supplemented) was lower than the mean of the control group (placebo). The ones in the right side indicate the opposite.

**Table 1 nutrients-13-04013-t001:** Bias risk of different aspects for the selected studies.

Study	Randomized (Selection Bias)	Blinding of Participants (Performance Bias)	Blinding of Outcome Assessment (Detection Bias)
Nieman et al., 2013 [32]	+	+	+
Shanely et al., 2014 [33]	+	+	+
Parsaie et al., 2019 [34]	+	+	+
Ashtary-Larky et al., 2020 [35]	+	+	+
Pilch et al., 2020 [36]	+	−	−
Żebrowska et al., 2020 [37]	−	+	+

(+) indicates low bias risk and (−) indicates high bias risk.

**Table 2 nutrients-13-04013-t002:** General characteristics of the selected studies.

**Athletic level of participants**	Young men/High school	2 studies: Pilch et al., 2020 [36] Shanely et al., 2014 [33]
	Resistance-trained men/Male runners	2 studies: Ashtary-Larky et al., 2020 [35] Żebrowska et al., 2020 [37]
	Male soccer players	1 study: Parsaie et al., 2019 [34]
	NASCAR pit crew	1 study: Nieman et al., 2013 [32]
**Age of participants**	<20 years old	1 study: Shanely et al., 2014 [33]
	20–30 years old	4 studies: Nieman et al., 2013 [32] Parsaie et al., 2019 [34] Ashtary-Larky et al., 2020 [35] Pilch et al., 2020 [36]
	30–40 years old	1 study: Żebrowska et al., 2020 [37]
**Vitamin D administration**	Ingestion	5 studies: Nieman et al., 2013 [32] Parsaie et al., 2019 [34] Pilch et al., 2020 [36] Shanely et al., 2014 [33] Żebrowska et al., 2020 [37]
	Injection	1 study: Ashtary-Larky et al., 2020 [35]
**Vitamin D dose** **(type)**	600 IU/day (vitD_2_)	1 study: Shanely et al., 2014 [33]
	2000 IU/week (vitD_3_)	1 study: Żebrowska et al., 2020 [37]
	3800 IU/day (vitD_2_)	1 study: Nieman et al., 2013 [32]
	50,000 IU/day (vitD_3_)	1 study: Parsaie et al., 2019 [34]
	300,000 IU injected in one dose (vitD_3_)	1 study: Ashtary-Larky et al., 2020 [35]
	Calculated from participant’s body mass using Singh and Bonham formula (vitD_3_)	1 study: Pilch et al., 2020 [36]
**Extent of supplementation**	3 weeks	1 study: Żebrowska et al., 2020 [37]
	4 weeks	1 study: Ashtary-Larky et al., 2020 [35]
	6 weeks	2 studies: Nieman et al., 2013 [32] Shanely et al., 2014 [33]
	8 weeks	1 study: Parsaie et al., 2019 [34]
	12 weeks	1 study: Pilch et al., 2020 [36]
**Exercise type**	Eccentric exercise	3 studies: Nieman et al., 2013 [32] Pilch et al., 2020 [36] Żebrowska et al., 2020 [37]
	Loughborough intermittent shuttle test	2 studies: Parsaie et al., 2019 [34] Shanely et al., 2014 [33]
	Acute resistance exercise (different presses and weights)	1 study: Ashtary-Larky et al., 2020 [35]

Abbreviations used: IU, international units; VitD_2_, ergocalciferol; VitD_3_, cholecalciferol.

**Table 3 nutrients-13-04013-t003:** Supplementation protocols.

Study	Circulating Vitamin D Levels (ng/mL) before the Study	Dose (IU)	Extent
Nieman et al., 2013 [32]	39 ± 2 (Experimental group *) 43 ± 3 (Placebo group)	3800 IU/day	6 weeks
Shanely et al., 2014 [33]	25 ± 2 (Experimental group *) 25 ± 2 (Placebo group)	600 IU/day	6 weeks
Parsaie et al., 2019 [34]	14 (Experimental group *) 15 (Placebo group)	50,000 IU/week	8 weeks
Ashtary-Larky et al., 2020 [35]	14 ± 3.9 (Experimental group *) 22 ± 5 (Placebo group)	300,000 IU injected	4 weeks
Pilch et al., 2020 [36]	30.5 ± 0.34 (OE group: Optimal levels of vitamin D at the beginning and supplemented during the intervention) 36.16 ± 4.4 (OC group: Optimal levels of vitamin D at the beginning and no supplemented during the intervention) 19 ± 1 (SE group: Suboptimal levels of vitamin D at the beginning and supplemented during the intervention) 14 ± 5 (SC group: Suboptimal levels of vitamin D at the beginning and no supplemented during the intervencion)	Calculated from participant’s body mass using Singh and Bonham formula.	12 weeks
Żebrowska et al., 2020 [37]	40 ± 8.8 (Experimental group *) 33.3 ± 3.4 (Placebo group)	2000 IU/day	3 weeks

(*) The experimental groups consume the corresponding dose of vitamin D indicated in the next column (“Dose (IU)”). Abbreviations used: IU, international units.

**Table 4 nutrients-13-04013-t004:** Sample size (participants) and main outcomes of the different studies.

Study	Sample Size	Analyzed Parameters (Time for Measurement)	Main Outcomes
Nieman et al., 2013 [32]	28	CK, LDH, Mb (immediately after exercise execution)	*↑*CK, *↑*LDH, *↑*Mb
Shanely et al., 2014 [33]	50	CK, LDH, Mb (immediately after exercise execution)	=CK, =LDH, =Mb
Parsaie et al., 2019 [34]	22	CK, LDH (immediately after exercise execution)	=CK, =LDH
Ashtary-Larky et al., 2020 [35]	14	CK, LDH (1 h after exercise execution)	=CK, =LDH
Pilch et al., 2020 [36]	60	CK, LDH, Mb (1 h after exercise execution)	*↓*CK, *↓*LDH, *↓*Mb *
Żebrowska et al., 2020 [37]	24	CK, LDH, Mb (1 h after exercise execution)	*↓*CK, *↓*LDH, *↓*Mb

Abbreviations and symbols used: CK, creatine kinase; LDH, lactate dehydrogenase; Mb, myoglobin, (↑) significant increased levels after intervention (negative outcome) in the supplemented group compared to placebo), (↓) significant decreased levels after intervention (positive outcome) in the supplemented group compared to the placebo group, (=) not significant differences after the intervention comparing experimental group to placebo. (*) Observed after 3 months of supplementation in the group with suboptimal levels of circulating vitamin D at the beginning of the study compared to a non-supplemented group. Articles include other outcomes not presented in the table.

**Table 5 nutrients-13-04013-t005:** The main limitations of the different studies of the meta-analysis.

Study	Limitations
Nieman et al., 2013 [32]	Physical activity consists in a very specific job, difficult to compare with other sport disciplines. Small sample size. Gender not specified (likely men). Supplementation with ergocalciferol.
Shanely et al., 2014 [33]	Information regarding the physical activity performed is limited, being complicated the comparison with other studies. Analytical method for vitamin D determination: chemiluminescent immunoassay. Performed only in men. Supplementation with ergocalciferol.
Parsaie et al., 2019 [34]	Analytical method for vitamin D determination: immunoassay. Small sample size. Gender not specified. Differences in baseline BMI between groups.
Ashtary-Larky et al., 2020 [35]	The exact type of endurance activity is not indicated. Analytical method for vitamin D determination: immunoassay (ELISA). Small sample size. Performed only in men. Short duration.One dose of vitamin D.
Pilch et al., 2020 [36]	Subjective control of the physical activity performed. The intensity is difficault to compare with other studies. Analytical method for vitamin D determination: immunoassay (ELISA). Performed only in men.
Żebrowska et al., 2020 [37]	Analytical method for vitamin D determination: no indicated. Small sample size. Performed only in men.

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
