# Peer review of "Effects of Vitamin D in Post-Exercise Muscle Recovery. A Systematic Review and Meta-Analysis"

_nutrients, 2021, doi:10.3390/nu13114013_

Round 1
Reviewer 1 Report
- Line 42, …vitamin D participates in the differentiation of monocytes to macrophages, ……..,
replace the “participates” option, involve
- Line 45, …. vitamin D participates in the modulation of inflammatory………
Replace the “participates” ,
- line 59-60, need to cite references
- line 65-68, need to cite references
- line 77, In addition, increased muscle damage goes together with…….”goes together ?
- line 94, ….upper and lower body segments. Which body segment?
- Line 95, The same authors [46]….. which author?
- References must be numbered in order of appearance in the text.
- Line 156, the table 1 title “(+) indicates low bias risk and (-) indicates high bias risk.” Looks like a footnote.
- Line 189, “Dose and extent for vitamin D supplementation are presented in Table 3.“ What does this phrase mean?
- Line 191-192, The study of Pilch et al (2020) was subdivided in 4 groups…., Need cite references.
- Line 191, ……in this study……, Is it mean in Pilch’s (2020) study? If yes, please rephrase the sentence.
- References numbered in text need to place before the punctuation. Ex, line 54, 72, 81, 92, 95, 154, 163, 205, 209, 213, 220, 237, 248, 259, 269, 272, 279, 283, 296, 304, 313, 337.
- Line 248, 259, “The study Pilch et al. 2020 [38] has been included several times as indicated before.” What does this phrase mean?
- Line 272, …..The same authors [46] published in a …….., which author?
- Reference 47 was omitted from the text.
Author Response
Dear Reviewer, please find attached the corrections to the suggestions that you have made. In the manuscript you can see the changes in red color.
REVIEWER-1
- Line 42, …vitamin D participates in the differentiation of monocytes to macrophages, ……..,replace the “participates” option, involve
ANSWER: Change has been done accordingly.
- Line 45, …. vitamin D participates in the modulation of inflammatory………Replace the “participates” ,
ANSWER: Change has been done accordingly.
- line 59-60, need to cite references
ANSWER: The references that support the statement are 13-15. These have been included in the text.
- line 65-68, need to cite references
ANSWER: The references that support the statement are 14 and 15. These have been included in the text.
- line 77, In addition, increased muscle damage goes together with…….”goes together ?
ANSWER: Change by “results in the” has been done accordingly.
- line 94, ….upper and lower body segments. Which body segment?
ANSWER: In sport sciences, it is more accurate to talk about upper and lower limbs as indicated in Reference 31. We have changed accordingly.
- Line 95, The same authors [46]….. which author?
ANSWER: There is mistake in the reference number. The statement can be inferred from Reference 31. The mistake has been corrected.
- References must be numbered in order of appearance in the text.
ANSWER: Once the point 7 has been corrected, the References are in the right order of appearance in the text.
- Line 156, the table 1 title “(+) indicates low bias risk and (-) indicates high bias risk.” Looks like a footnote.
ANSWER: The sentence has been placed as a footnote, as suggested by the Reviewer.
- Line 189, “Dose and extent for vitamin D supplementation are presented in Table 3.“ What does this phrase mean?
ANSWER: This refers to amount and time of supplementation. The sentence has been changed accordingly.
- Line 191-192, The study of Pilch et al (2020) was subdivided in 4 groups…., Need cite references.
ANSWER: This is Reference 38. This has been indicated in the text.
- Line 191, ……in this study……, Is it mean in Pilch’s (2020) study? If yes, please rephrase the sentence.
ANSWER: This refers to Reference 38. This has been indicated in the text.
- References numbered in text need to place before the punctuation. Ex, line 54, 72, 81, 92, 95, 154, 163, 205, 209, 213, 220, 237, 248, 259, 269, 272, 279, 283, 296, 304, 313, 337.
ANSWER: The place of the References has been changed as suggested in all sentences.
- Line 248, 259, “The study Pilch et al. 2020 [38] has been included several times as indicated before.” What does this phrase mean?
ANSWER: This refers to section 3.2 (lines 191-200). We have clarified this point referring to section 3.2.
- Line 272, …..The same authors [46] published in a …….., which author?
ANSWER: Authors of references 45 and 46 are form the same research group. We have indicated accordingly.
- Reference 47 was omitted from the text.
ANSWER: Reference 47 is cited now in the text (2nd paragraph of Discussion).
Reviewer 2 Report
Thank you ever so much as taking into account so many points, there is now considerably more details in there which can help the readers get a better feel for the evidence you are synthesising. I think there remains an issue that you make comments that the evidence you have looked at doesn't support. For example the penultimate sentence of the abstract, talking about more studies one could do. It is good discuss further work in regards to the fact that the trials were done poorly but it must be tempered by the fact that you have shown there is no effect. However you could conclude you can't conclude anything at all given the limitations of the experiments, the heterogeneity and I also don't agree the groups were well matched. I think some of the starting vitamin Ds are quite different.
I also think your introduction is interesting when you say that there are multiple factors affecting CK rises for example and the interindividual variation so do you think the controls are rigorous enough? Also, may be my ignorance, but how do you distinguish a rise in CK associated with normal/healthy response to training from muscle damage?
Therefore altogether much better but I still think there is a lot of discussion of what vitamin D does but perhaps not enough focus on what you have shown which is probably poor studies showing no effect.
Author Response
Dear Reviewer, please find attached the corrections to the suggestions that you have made. In the manuscript, you can see the changes in blue color.
REVIEWER-2
Thank you ever so much as taking into account so many points, there is now considerably more details in there which can help the readers get a better feel for the evidence you are synthesising. I think there remains an issue that you make comments that the evidence you have looked at doesn't support. For example the penultimate sentence of the abstract, talking about more studies one could do. It is good discuss further work in regards to the fact that the trials were done poorly but it must be tempered by the fact that you have shown there is no effect. However you could conclude you can't conclude anything at all given the limitations of the experiments, the heterogeneity and I also don't agree the groups were well matched. I think some of the starting vitamin Ds are quite different.
ANSWER: We agree that the selected studies have limitations that have been collected in the manuscript in a table. We have indicated this point in the Abstract.
I also think your introduction is interesting when you say that there are multiple factors affecting CK rises for example and the interindividual variation so do you think the controls are rigorous enough? Also, may be my ignorance, but how do you distinguish a rise in CK associated with normal/healthy response to training from muscle damage?
ANSWER: CK is a marker of muscular damage and this damage can occur after a high intensity training and derived from some pathologies. Additional medical probes are necessary to determine the origin of increased levels of circulating CK.
Therefore altogether much better but I still think there is a lot of discussion of what vitamin D does but perhaps not enough focus on what you have shown which is probably poor studies showing no effect.
ANSWER: We guess that the correct thing is focus the Discussion in the selected studies, as we did. The problem is that studies have a poor design and conclusions cannot be reached accordingly.
This manuscript is a resubmission of an earlier submission. The following is a list of the peer review reports and author responses from that submission.
Round 1
Reviewer 1 Report
thank you for the opportunity to review your work, it is clear you have done a lot. In many ways I could follow but there are multiple grammar and spelling mistakes.
Intro: Enzyme isoforms - these are not routinely available and not sure what the relevance is of this part of the discussion. Mainly I am struck by the absence of reviewing previous reviews on the subject of vitamin D supplementation (which nicely show there is no effect too e.g. cochrane).
Some example spelling mistakes e.g. Page 2 - proliferation, Myocyte
Methods: why athletes? Not sure that was explained in the introduction, why would their response be different. May be I can see now why you mention the enzymes. However the issue that now becomes clear is that you may have discussed isoenzymes but not the variation in method types e.g. direction and cofactors so how can you/did you compare assay results across studies as eznyme measurement is not standardized? You don't measure vitamin D analysis issues i.e. the fact that that international editors have stipulated steroid hormones should be measured by mas spec if being reported and therefore I can not tell if you are reporting immunoassay results, mas spec or a mixture of both. You don't mention the type of vitamin D supplemented either.
discussion - you now mention that others have done meta analysis and reviews, I think some of this should be in the introduction. you do not discuss the limitations of your studies, the markers, the techniques etc so don't appear to really discuss your study in any detail. You don't discuss I2 results etc which would suggest a large degree of heterogeneity. Your conclusion sounds too positive too. Effectively yours and others work has shown no effect and cochrane etc have called for all to stop wasting resources on further work on vitamin D as no one has ever demonstrated a robust effect despite a staggering number of trials therefore it is not felt to be due to the fact that further trials need to be done, the call is to stop completely. Your findings have agreed with that so it would be more fitting to agree with the international consensus that there is no role in vitamin D supplementation outside of rickets and osteomalacia.
All in all the work isn't novel as although you may have looked at a subtly different marker of muscle function you do not discuss if that is valid and it is well known vitamin D doesn't do anything so not really a surprise it was negative. I also think your conclusion does not match your findings or the published literature. Your review of the subject, methods, markers etc is not expert enough nor demonstrates critical appraisal skills.
Author Response
REVIEWER 1
Thank you for the opportunity to review your work, it is clear you have done a lot. In many ways I could follow but there are multiple grammar and spelling mistakes.
We have sent a lecturer from the Faculty of Translation and Interpreting to review the manuscript. However, if, at this stage, you feel that further implementation of the English language is required, we could apply to the English editing service offered by the Editorial Board, at the appropriate cost, and the manuscript would be revised according to your suggestion. In this regard, I look forward to hearing from you.
Intro: Enzyme isoforms - these are not routinely available and not sure what the relevance is of this part of the discussion. Mainly I am struck by the absence of reviewing previous reviews on the subject of vitamin D supplementation (which nicely show there is no effect too e.g. cochrane).
ANSWER: We have indicated in the Introduction that the enzyme isoforms used to determine muscle damage are specific from muscular tissue. The advantage is that they can be detected in a routine blood analysis. On the other hand, no reviews regarding specifically the topic of post-exercise muscle recovery have been found.
Some example spelling mistakes e.g. Page 2 - proliferation, Myocyte
ANSWER: The typing mistakes have been corrected.
Methods: why athletes? Not sure that was explained in the introduction, why would their response be different. May be I can see now why you mention the enzymes. However the issue that now becomes clear is that you may have discussed isoenzymes but not the variation in method types e.g. direction and cofactors so how can you/did you compare assay results across studies as enzyme measurement is not standardized? You don't measure vitamin D analysis issues i.e. the fact that that international editors have stipulated steroid hormones should be measured by mas spec if being reported and therefore I can not tell if you are reporting immunoassay results, mas spec or a mixture of both. You don't mention the type of vitamin D supplemented either.
ANSWER: Athletes is used to indicate in general sportsmen and sportswomen. We used athletes because is shorter.
In addition, we choose variations in circulating muscle isoenzymes to identify the existence of muscular damage. The muscle biomarkers can be detected in a routine blood analysis. This is the widely accepted standard procedure to detect muscle damage in all sports disciplines.
Furthermore, the objective of the meta-analysis is not to compare the reliability and sensitivity of the different analytical methods. We only payed attention that vitamin D levels were reported in the different publications selected as well as an increase in the levels of muscle biomarkers compared to a placebo group. Of course, into each study, the analytical method was the same for the experimental as well as for the placebo group. For this reason, results can be compared into each particular study.
Finally: The type of vitamin D is indicated in Table 2.
Discussion: - you now mention that others have done meta analysis and reviews, I think some of this should be in the introduction. You do not discuss the limitations of your studies, the markers, the techniques etc so don't appear to really discuss your study in any detail. You don't discuss I2 results etc which would suggest a large degree of heterogeneity. Your conclusion sounds too positive too. Effectively yours and others work has shown no effect and cochrane etc have called for all to stop wasting resources on further work on vitamin D as no one has ever demonstrated a robust effect despite a staggering number of trials therefore it is not felt to be due to the fact that further trials need to be done, the call is to stop completely. Your findings have agreed with that so it would be more fitting to agree with the international consensus that there is no role in vitamin D supplementation outside of rickets and osteomalacia.
ANSWER: Only one meta-analysis is cited in the Discussion (Tomlinson et al., 2015). This report analyzed the role of vitamin D supplementation on muscle strength, indicated positive effects. However, they do not present data regarding muscle recovery. We have moved the paragraph that cites this study to the Introduction as suggested by the Reviewer.
Regarding the limitations, we have included this point in the Discussion. We believe that the main limitation of these studies is that the anti-inflamamtory action exerted by vitamin D (well documented by several reports) is not clearly monitored analysing circulating levels of specific muscle markers. Otherwise said, the efficiency of vitamin D supplementation decreasing immunological markers of inflammation cannot be totally extended to circulating biomarkers of muscle damage. To circumvent this problem, cytokine determinations should be included in studies regarding muscle recovery. In this line, parameters determined in regular blood analysis do not seem to be useful to monitor muscle recovery.
The heterogeneity reported by I2 results is indicated now at the end of Discussion and in the “Conclusions” section.
Regarding Cochrane position, we did this meta-analysis because we did not found a Cochrane position regarding vitamin D and muscle recovery after PubMed and other database consultation. In addition, Cochrane web site did not claim to stop wasting resources on vitamin D and muscle recovery. As we have said in the limitations, the anti-inflammatory effect of vitamin D should be considered in future work by addressing cytokine determinations. The data obtained from regular blood analysis seem to be incomplete.
En general, el trabajo no es novedoso, ya que aunque es posible que haya observado un marcador sutilmente diferente de la función muscular, no discute si eso es válido y es bien sabido que la vitamina D no hace nada, por lo que no es realmente una sorpresa que fuera negativo. También creo que su conclusión no coincide con sus hallazgos o la literatura publicada. Su revisión del tema, métodos, marcadores, etc. no es lo suficientemente experta ni demuestra habilidades críticas de evaluación.
RESPUESTA: La novedad del metaanálisis es que los biomarcadores musculares circulantes no permiten monitorizar adecuadamente el proceso de recuperación. La determinación de citoquinas y un protocolo de actividad física de consenso deben realizarse en futuras investigaciones.
Reviewer 2 Report
The limitation of this review article is that only six relevant studies were analyzed. There are some suggestions for authors.
- English writing should be improved by a native English speaker. Ex, in line 43, the sentence “…vitamin D participants in the…,” in line 46, “…vitamin D participates in the modulation…”.
- In line 60, please check “skeletal miocytes… “
- In line 64, ALT is not a biomarker of muscle damage, it is the best-known of liver injury biomarker.
- The introduction section, paragraph 3 which is confused, because the delayed onset of muscle soreness is occur on skeletal muscle. However, Mb, CK-MB, AST, LDH are the biomarkers after myocardial infarction onset.
- In line 187, “In this study, participants were divided at the beginning of the…” the sentence is difficult to read. Isn’t this article a systemic review article?
- Please consistent the embedded citations in the text. Ex, Beaudart et al [27]; Close et al. [43].
Author Response
REVIEWER-2
The limitation of this review article is that only six relevant studies were analyzed. There are some suggestions for authors.
English writing should be improved by a native English speaker. Ex, in line 43, the sentence “…vitamin D participants in the…,” in line 46, “…vitamin D participates in the modulation…”.
ANSWER: We have sent a lecturer from the Faculty of Translation and Interpreting to review the manuscript. However, if, at this stage, you feel that further implementation of the English language is required, we could apply to the English editing service offered by the Editorial Board, at the appropriate cost, and the manuscript would be revised according to your suggestion. In this regard, I look forward to hearing from you.
In line 60, please check “skeletal miocytes… “
ANSWER: This typing error has been corrected.
In line 64, ALT is not a biomarker of muscle damage, it is the best-known of liver injury biomarker.
ANSWER: Many reports consider ALT as a muscle biomarker. We agree with the Reviewer that ALT is more specific for liver damage. This is why we have indicated this point in the 3rd paragraph of Introduction.
The introduction section, paragraph 3 which is confused, because the delayed onset of muscle soreness is occur on skeletal muscle. However, Mb, CK-MB, AST, LDH are the biomarkers after myocardial infarction onset.
ANSWER: This is true, but in this particular case, the cardiac isoforms (i.e. CK) are determined. In our meta-analysis we refer only to skeletal muscle isoforms.
In line 187, “In this study, participants were divided at the beginning of the…” the sentence is difficult to read. Isn’t this article a systemic review article?
ANSWER: The study of Pilch et al is not a systematic review, it is a randomized control trial. In this study, participants were divided at the beginning of the study in 2 groups: participants with optimal levels of circulating vitamin D and participants with suboptimal levels of vitamin D. Then, each group was divided in 2 groups: experimental or supplemented and control or non-supplemented. This gives a total of 4 groups as indicated in the Results.
Por favor, haga coherentes las citas incrustadas en el texto. Ex, Beaudart et al [27]; Close et al. [43].
RESPUESTA: Suponemos que el Revisor se refiere a mantener el mismo formato para las citas en el texto. Hemos cambiado en consecuencia.
Round 2
Reviewer 1 Report
Thank you ever so much for addressing my comments. Looking at the English there are still some subtle errors but not enough to stop me reading it fluently therefore if the editorial team is happy to tweak some minor spelling and grammar issues etc I would have no problem with the written work. It is in fact very good and the issues are subtle.
I wonder if some of my issues were perhaps misunderstood. The question about athletes may be my ignorance but I appreciate athlete is a shorter term than sportsman or woman but why did you exclude non sportspeople and how do define athlete? I am not familiar with sports science so there may be a definition of what an athlete is but you say you did not exclude any one based on a whole host of factors including the type of effort etc so is someone who does a short walk every day an athlete? Are the NASCAR pitline crew athletes?
re vit D and Cochrane it is mainly in editorials where you see the calls to stop all further studies. Multiple guidelines etc have been attacked by the authors of the Cochran’s review who have been working for years to get papers removed from the literature using fictional data and reviews which cite them in a huge scandal that resulted in Sato’s suicide. https://www.saragironicarnevale.com/science-magazine-tide-of-lies I think it would be pertinent to mention this even in passing that the use of widespread testing and treatment in humans is due to research fraud. This is why journal editors have become more circumspect with papers on vitamin D or at least should be. There is also a general impression along the same lines in regards to anti-oxidants and reactive oxygen species. For many journals their author guidelines include that they will not accept any articles on antioxidant status so I should be cautious of mentioning this in general, I don’t know if you know the half life of a reactive oxygen species but it is definite that we can never hope to measure a change. Again I would see cochrane review on vitamin Aand E supplementation (hypothesis being antioxidants) that demonstrates increased mortality. Anyway this journal itself may be more open to articles including hypotheses that vitamin d or antioxidant etc may be helpful but to present a more balanced argument you should present the converse opinion that this is not a widely supported hypothesis at all. Basically your introduction is not an expert introduction I feel, it reads as biased with no rigorous critical review of the papers you cite. Which is also apparent in the discussion where you state that you don’t believe the increase in muscle enzymes in the NASCAR pit crew post vit D supplementation. If there are so many problems with interpreting this study then surely some of those limitations apply to the other 5 studies too, singling out this one made me feel you didn’t like this one in particular as it was most against your hypothesis. Basically you need to tear into each of the 6 studies and point all the limitations in a less biased way.
in regards to measurement, again thank you for adding the types of vitamin D, but I don’t think you appreciate the point. https://doi.org/10.1177%2F0004563218796858 Here is a nice quick editorial that summarises some of the issues. At the very least you need to tell us how each study measured vit D and what the cross reactivity of D2 is in them. Our hospital assay has a recovery of only 80% for D2 hence why only mass spec methods are meant to be used in research. This is one of the issues you need to discuss.
Back to the supplements, another thing lacking in your critical review is what you think of the dosing regimes. Is d2 acceptable? That are the pharmacodynamic and kinetic differences? We have already mentioned the analytical differences. If you look at the starting vitamin D in 34, 37, 38, and 39 do you think the placebo and treatment groups we well matched. The SD are super different e.g. in Pilch et al. Are these well matched?
Also you don’t comment on the mega dosage of ashtary etc and how this study showed no effect at all. If your hypothesis is correct would you not expect more vit D to do more and quicker?
effectively I think you have done a lot of work and you have presented it well however it reads as very biased and not particularly expert. I am certain you understand the topic well but I do not understand it with you. In the introduction to improve you should employ your knowledge and synthesise the experimental and clinical studies to provide a much more balanced approach. You mention anti-inflammatory but I know if no clinical use of vit D as an anti-inflammatory therefore do experimental associations actually equate to a meaningful effect? You talk a lot about muscle enzymes, this is perhaps unnecessary particularly as you don’t tell us which tests were actually performed and most people would appreciate ALT is a liver marker, which one of the studies used ALT by the way, is it relevant? I mentioned First feedback the method but it is important the analytical method but I appreciate at least each study used the same ( but do you actually tell us that in results?) as the enzyme set up will effect the performance of the marker, the reference range etc…
In the methods and results you need to tell us the vit D and enzyme methods and, you will know more about this than me, have you presented enough information about the exercise etc for knowledgeable audience to appreciate if it was a robust test or not? It seems sparse to me but I am not a sports physiologist.
This then means you can give us a much better review in your discussion. Firstly you have shown no effect, however heterogeneity exists and the experiments were poor due to numerous reasons which you explain to us. The studies are also small and you could very slightly hypothesise that may be the enzymes wrong markers and cytokines may be preferable, but cautious here as you have no evidence for that. You can then conclude that it very nicely supports the prevailing opinion that there is no role for fat soluble vitamin supplementation outside of clinically manifest deficiency syndromes with proven hypovitaminosis. (The current clinical thinking is that vit d measurement and supplementation is extremely expensive, a waste of time and money and in the cases of vitamins A and E harmful)
hope that makes sense but without mentioning the problems with the literature, the measurement, supplementation, experimental design etc it reads as biased and too superficial. Of course you may think I am too biased against vitamin D but you failed to convince me I am wrong and that your suggestion for more work is justified. At the very least if you can critically appraise those studies more then you can take us with you if You are suggesting the negative effect is spurious. Hope this helps.
Author Response
REVIEWER-1 (round 2)
1) Thank you ever so much for addressing my comments. Looking at the English there are still some subtle errors but not enough to stop me reading it fluently therefore if the editorial team is happy to tweak some minor spelling and grammar issues etc I would have no problem with the written work. It is in fact very good and the issues are subtle.
ANSWER: We appreciate the comment and we will ask to the Editorial Board to perform a final revision of the English before publication.
2) I wonder if some of my issues were perhaps misunderstood. The question about athletes may be my ignorance but I appreciate athlete is a shorter term than sportsman or woman but why did you exclude non sportspeople and how do define athlete? I am not familiar with sports science so there may be a definition of what an athlete is but you say you did not exclude any one based on a whole host of factors including the type of effort etc so is someone who does a short walk every day an athlete? Are the NASCAR pitline crew athletes?
ANSWER: We are working in Sport Nutrition and we are interested in all aspects related to this topic. Indeed, recovery after exercise execution is one of the aspects we are interested. Therefore, the population segment considered as sportsmen/sportswomen is people that develop a planned activity regarding intensity, volume and frequency, performed in a professional o semi-professional way. In this context, people that does a short walk is not an athlete. We have explained the term “athlete” at the end of Section 2.2. Regarding the NASCAR pitline crew athletes, the title of the publication indicates clearly that they were sportsmen performing an eccentric exercise: “Nieman, D.C.; Gillitt, N.D.; Shanely, R.A.; Dew, D.; Meaney, M.P.; Luo, B. Vitamin D2 supplementation amplifies eccentric exercise-induced muscle damage in NASCAR Pit Crew athletes”. We selected this article according the “Title” and the “Abstract”. However, when we read it in more detail we observed some gaps that we have pointed in the last paragraph of “Discussion”. We believe that this information could be helpful for readers of the manuscript.
3) Vit D and Cochrane it is mainly in editorials where you see the calls to stop all further studies. Multiple guidelines etc have been attacked by the authors of the Cochran’s review who have been working for years to get papers removed from the literature using fictional data and reviews which cite them in a huge scandal that resulted in Sato’s suicide. https://www.saragironicarnevale.com/science-magazine-tide-of-lies I think it would be pertinent to mention this even in passing that the use of widespread testing and treatment in humans is due to research fraud. This is why journal editors have become more circumspect with papers on vitamin D or at least should be.
ANSWER: Sato’s research was focused in the role of vitamin D in reducing bone fractures. This was one of the biggest frauds in science. We guess that many researchers are doing their work under good standard practices. We cannot imply that everybody is guilty because one person is out of the ethics when publishing. In any case, we have taken into account this appreciation and commented accordingly (Section 2.1).
4) There is also a general impression along the same lines in regards to anti-oxidants and reactive oxygen species. For many journals their author guidelines include that they will not accept any articles on antioxidant status so I should be cautious of mentioning this in general, I don’t know if you know the half life of a reactive oxygen species but it is definite that we can never hope to measure a change. Again I would see cochrane review on vitamin A and E supplementation (hypothesis being antioxidants) that demonstrates increased mortality. Anyway this journal itself may be more open to articles including hypotheses that vitamin D or antioxidant etc may be helpful but to present a more balanced argument you should present the converse opinion that this is not a widely supported hypothesis at all.
ANSWER: We agree with the Reviewer that anti-oxidants and reactive oxygen species are a very debated issue. Reactive oxygen species have a very short half-life (less than one second). For this reason, we prefer data obtained from the measurement of oxidized organic molecules, such as protein carbonyls, malondialdehyde (final end-product of lipid peroxidation) or oxoguanine (typical product from DNA oxidation). These oxidation end-products have a longer half-live and are usually determined in many publications, even in high impact journals. Intracellular antioxidant, need the production of some free radicals to induce the corresponding coding genes. For that reason, high dose of certain antioxidants (i.e. vitamins A and E) could avoid free radical formation and thereby the induction of endogenous antioxidant enzymes, that are more specific to mitigate oxidative damage. For that reason, antioxidant supplementation is not healthy at high doses. Altogether, the best antioxidant is this one that allows antioxidant enzyme gene expression at the same time that acts such as an exogenous antioxidant (complementary action: exogenous response + adaptive response). We have pointed this controversial point in more detail in 4th paragraph of the Introduction.
5) Basically your introduction is not an expert introduction I feel, it reads as biased with no rigorous critical review of the papers you cite. Which is also apparent in the discussion where you state that you don’t believe the increase in muscle enzymes in the NASCAR pit crew post vit D supplementation. If there are so many problems with interpreting this study then surely some of those limitations apply to the other 5 studies too, singling out this one made me feel you didn’t like this one in particular as it was most against your hypothesis. Basically you need to tear into each of the 6 studies and point all the limitations in a less biased way.
ANSWER: The goal of the Introduction is to present the role of vitamin D in several muscle functions determined in different publications, including strength, pain, balance, lesions. We also present data regarding the well documented role of vitamin D in modulation of inflammation. Finally, we introduce the classical muscle markers that are widely used to determine muscle damage: CK, LDH and Mb. On the other hand, Discussion is devoted to the 6 selected studies of the meta-analysis. We guess that is a good idea to indicate the limitations of each study (see new Table 5).
6) In regards to measurement, again thank you for adding the types of vitamin D, but I don’t think you appreciate the point. https://doi.org/10.1177%2F0004563218796858 Here is a nice quick editorial that summarises some of the issues. At the very least you need to tell us how each study measured vit D and what the cross reactivity of D2 is in them. Our hospital assay has a recovery of only 80% for D2 hence why only mass spec methods are meant to be used in research. This is one of the issues you need to discuss.
ANSWER: We agree that liquid chromatography-mass spectrometry are the gold standard for measuring 25(OH)D. Immunoassay has a lower quality and specificity. We think that this information could be useful for readers. We have considered this point as a limitation of each study and included in the new Table 5 and the last paragraph of Discussion.
7) Back to the supplements, another thing lacking in your critical review is what you think of the dosing regimes. Is D2 acceptable? That are the pharmacodynamic and kinetic differences? We have already mentioned the analytical differences. If you look at the starting vitamin D in 34, 37, 38, and 39 do you think the placebo and treatment groups were well matched. The SD are super different e.g. in Pilch et al. Are these well matched?
ANSWER: It seems that vitamin D3 (cholecalciferol) is more efficacious at raising serum active vitamin D concentrations than vitamin D2 (ergocalciferol). These differences seem to be related to differences in hydroxylation and affinity for liver enzymes and receptors. As mentioned before, these are limitations of the studies indicated in the new Table 5, the last paragraph of Discussion and new Reference 50. On the other hand, we think that the studies are well matched, but we raised some concerns for references 36 and 39. This is now indicated in Section 2.4.
8) Also you don’t comment on the mega dosage of Ashtary etc and how this study showed no effect at all. If your hypothesis is correct would you not expect more vit D to do more and quicker?
ANSWER: We agree that Ashtary-Larky et al used one megadose only once with no post-exercise changes in CK and LDH circulating levels. The dose to be administered and the length of supplementation is a pending question for future research. We have included this point in the 3th paragraph of Discussion.
9) Effectively I think you have done a lot of work and you have presented it well however it reads as very biased and not particularly expert. I am certain you understand the topic well but I do not understand it with you. In the introduction to improve you should employ your knowledge and synthesise the experimental and clinical studies to provide a much more balanced approach. You mention anti-inflammatory but I know if no clinical use of vit D as an anti-inflammatory therefore do experimental associations actually equate to a meaningful effect? You talk a lot about muscle enzymes, this is perhaps unnecessary particularly as you don’t tell us which tests were actually performed and most people would appreciate ALT is a liver marker, which one of the studies used ALT by the way, is it relevant? I mentioned First feedback the method but it is important the analytical method but I appreciate at least each study used the same ( but do you actually tell us that in results?) as the enzyme set up will effect the performance of the marker, the reference range etc…
ANSWER: Introduction is just to present some data regarding the role of vitamin D in modulation of inflammation. This is a well stablished topic according to the literature. In addition, we have deleted ALT and AST from the list of muscle biomarkers. In fact, in the selected publications of the meta-analysis, these enzymes are not determined. Finally, the analytical method is indicated as a limitation in the new Table 5.
10) In the methods and results you need to tell us the vit D and enzyme methods and, you will know more about this than me, have you presented enough information about the exercise etc for knowledgeable audience to appreciate if it was a robust test or not? It seems sparse to me but I am not a sports physiologist.
ANSWER: We have included this information in a new paragraph between Tables 2 and 3.
11) This then means you can give us a much better review in your discussion. Firstly you have shown no effect, however heterogeneity exists and the experiments were poor due to numerous reasons which you explain to us. The studies are also small and you could very slightly hypothesise that may be the enzymes wrong markers and cytokines may be preferable, but cautious here as you have no evidence for that. You can then conclude that it very nicely supports the prevailing opinion that there is no role for fat soluble vitamin supplementation outside of clinically manifest deficiency syndromes with proven hypovitaminosis. (The current clinical thinking is that vit D measurement and supplementation is extremely expensive, a waste of time and money and in the cases of vitamins A and E harmful).
ANSWER: Determination of inflammatory cytokines to monitor post-exercise recovery is a proposal. As we said at the end of the 3th paragraph of the Discussion, we need further research to address this point. We think that as scientists we have to propose new ways and stimulate research by other expert groups.
12) Hope that makes sense but without mentioning the problems with the literature, the measurement, supplementation, experimental design etc it reads as biased and too superficial. Of course you may think I am too biased against vitamin D but you failed to convince me I am wrong and that your suggestion for more work is justified. At the very least if you can critically appraise those studies more then you can take us with you if You are suggesting the negative effect is spurious. Hope this helps.
ANSWER: The modifications made in the Discussion (including new Table 5) and suggested by the Reviewer resulted in more critical analysis. We guess that the meta-analysis is more balanced and fits better with the actual state of the art.
Reviewer 2 Report
I have no objections to its publications.
Author Response
REVIEWER-2 (round 2)
I have no objections to its publications.
ANSWER: Thanks a lot. We hope to solve the remaining points raised by the other Reviewer in a satisfactory way.